# Towards a Better Understanding of Endometriosis-Related Infertility: A Review on How Endometriosis Affects Endometrial Receptivity

**DOI:** 10.3390/biom13030430

**Published:** 2023-02-24

**Authors:** Jing Shan, Da-Jin Li, Xiao-Qiu Wang

**Affiliations:** 1Obstetrics and Gynecology Hospital of Fudan University, Shanghai 200011, China; 2Department of Obstetrics and Gynecology, Hainan Medical College Affiliated Hospital, Haikou 571100, China

**Keywords:** endometriosis, endometrial receptivity, inflammation, immunoregulatory, epigenetic, glycosylation, microRNA

## Abstract

Endometriosis is the most common cause of infertility. Endometrial receptivity has been suggested to contribute to infertility and poor reproductive outcomes in affected women. Even though experimental and clinical data suggest that the endometrium differs in women with endometriosis, the pathogenesis of impaired endometrial receptivity remains incomplete. Therefore, this review summarizes the potential mechanisms that affect endometrial function and contribute to implantation failure. Contemporary data regarding hormone imbalance, inflammation, and immunoregulatory dysfunction will be reviewed here. In addition, genetic, epigenetic, glycosylation, metabolism and microRNA in endometriosis-related infertility/subfertility will be summarized. We provide a brief discussion and perspectives on their future clinical implications in the diagnosis and therapy to improve endometrial function in affected women.

## 1. Introduction

Endometriosis (EMS) is an estrogen-dependent disease characterized by the implantation of endometrial glands and stroma outside the uterine cavity. About 30–50% of women with EMS are infertile, while 20–50% of infertile women are diagnosed with EMS. Endometrial receptivity is reported to decrease in patients with clinically diagnosed EMS [1]. However, the mechanism by which EMS impairs endometrial receptivity is not fully understood. Recently, studies have found changes in the endometrial gene expression, sex hormone receptors, and cell adhesion molecules in women with EMS, suggesting that inherent abnormalities in endometrial function are present in women with EMS [2].

A successful pregnancy depends on the interactions between the embryo and endometrium in the human reproductive system. The human endometrium is a complex and highly active organ in the female reproductive function. The endometrium participates in receiving the embryo, promoting embryo implantation and supporting embryonic growth, decidualization, and development until the placenta develops. An impaired endometrial function may disrupt the window of implantation (WOI) and interrupt implantation, which can lead to infertility. Therefore, identifying the mechanisms that affect endometrial function and contribute to implantation failure is essential for guiding therapeutic strategies. Thus, this review aims to summarize and analyze the evidence for an “endometrial factor” causing EMS-related infertility/subfertility and explore the potential interventions to improve endometrial function in affected women based on the reasons mentioned earlier.

## 2. Methods

A comprehensive search of Embase, Medline and Cochrane Central databases was conducted for the literature up to December 2022. The search strategy used the following keywords: “endometriosis”, “ovarian endometriosis cysts”, “infertility”, “endometrial receptivity”, “embryo implantation”, “progesterone resistance”, “estrogen dominance”, “inflammation”, “immunoregulatory”, “macrophages”, “natural killer cells”, “dendritic cells”, “epigenetic”, “glycosylation”, “metabolites”, “miRNA”, “treatment”, “surgery”, “immunomodulation”, and “assisted reproductive technology”. The search was limited to full documents published in peer-reviewed journals in English.

## 3. Endometrial Molecular Differences in Endometriosis for Implantation

During WOI, the endometrium undergoes a series of molecular changes to accommodate embryo implantation. Several studies have reported the gene profile and protein expression in the eutopic endometrium of women with different stages of EMS during the secretory phase [3]. The results showed that the aberrant expression of genes and proteins resulted in the disruption of several cellular processes, i.e., cell cycle, cell apoptosis, cell adhesion/migration, immune/inflammation regulation, energy production, signal transduction, and decidualization, leading to impaired endometrial receptivity in the endometrium of women with EMS (Table 1). Giudice et al. observed the occurrence of molecular dysregulation of progesterone target genes during the secretory phase of endometrium in women with EMS, which explains the attenuated progesterone responsiveness [4]. Tayade et al. displayed differential expressions of immune-inflammation genes in the eutopic endometrium of EMS patients compared with control samples [5]. Aberrantly expressed gene products with important roles in the establishment of receptivity include transcription factors (e.g., HOXA10 and HOXA11) [6,7], adhesion molecules (e.g., integrin αvβ3 and CD44) [8,9], cytokines (e.g., leukemia inhibitory factor (LIF) and epidermal growth factor (EGF)), and regulated steroid hormone metabolizing enzymes (e.g., aromatase,17β-hydroxysteroid dehydrogenase). Therefore, this review summarizes the alterations in the endometrial milieu during WOI in women with EMS. We can conclude that there is an impaired endometrial function in patients with EMS.

## 4. Dysregulation of Progesterone and Estrogen Signaling

The human endometrium is a complicated and dynamic tissue comprised of epithelial (lumen and glandular epithelial cells), stromal, immune, and vascular cells. These cells are targets of ovarian steroid hormones, including estrogen (mainly 17β-estradiol) and progesterone. Endometrial paracrine signaling is needed for successful embryo implantation. Ovarian steroid hormones control uterine physiology by binding to the estrogen receptor (ER/ESR) and progesterone receptor (PR/PGR) to regulate gene transcription. In the normal endometrium, progesterone and estrogen signaling work together to inhibit epithelial proliferation and facilitate the transition to an embryo-receptive state during WOI. As a response to the implanted embryo, the surrounding stromal cells undergo decidualization to sustain the growth and invasion of the embryo. However, the dysfunction of the progesterone and estrogen signaling in EMS patients results in progesterone resistance and estrogen dominance [14].

### 4.1. Progesterone Resistance

Progesterone (P4) is a key hormone regulator responsible for preparing the uterus for implantation and also for establishing and maintaining a pregnancy. The uterine response to progesterone depends on the PR. Progesterone modulates the endometrium through two receptor subtypes (PR-A and PR-B), which act primarily as ligand-activated transcription factors. PR-specific knockout in endometrial epithelial cells can lead to pregnancy failure because of unlimited estrogen-induced epithelial cell proliferation and impaired stromal decidualization [15]. Female mice which lack PRA and PRB exhibit reproductive defects. Therefore, epithelial PR expression is crucial for embryo implantation, suppression of estrogen-induced epithelial proliferation, and stromal decidualization. If the endometrium does not respond well to progesterone exposure, this is termed progesterone resistance. In EMS, it is manifested as a failure to induce PGR activation [16]. The expression of PGR is associated with multiple transcriptional pathways. AT-rich interacting domain protein 1A (ARID1A) is a chromatin-remodeling complex protein that plays a key role in steroid hormone signaling. It has been reported that PGR may be a target gene of ARID1A. The expression level of ARID1A significantly decreased in the endometrium of infertile patients with EMS compared to those without EMS [17].

The P4-induced Ihh-COUP-TFII signaling axis mediates epithelial–stromal interactions that are crucial for embryo implantation and stromal decidualization. The Indian Hedgehog (Ihh) ligands bind to patched receptors (Ptch1 and Ptch2) and subsequently increase the expression of COUP-TFII in the stromal cell. Chicken ovalbumin upstream promoter-transcription factor II (COUP-TFII) is an orphan nuclear receptor that modulates uterine physiology for proper embryo implantation and decidualization. Ihh mediates uterine receptivity and decidualization by activating multiple signal pathways (e.g., Ihh-Ptch1-COUP-TFII- Hand2-Bmp2 and Wnt4 signaling). However, Ihh signaling is inhibited in EMS patients [18]. Disturbance of upstream molecular signaling in the endometrium, such as KRAS activation and overexpressed SIRT1/BCL6 [19], may disrupt the endometrial Ihh signaling. The Ihh signaling axis is a pivotal mediator of PGR in the uterus; therefore, impaired Ihh signaling may lead to progesterone resistance in EMS.

Heart- and neural crest derivatives-expressed protein 2 (Hand2) is a key mediator between active progesterone signaling and inhibition of estrogen-induced epithelial cell proliferation. Fibroblast growth factors (FGFs) can activate the extracellular signal-regulated kinase (ERK) pathway by binding to their receptors FGFRs, which is critical for promoting the expression of ERα. Up-regulation of ERα in endometrial epithelial cells promotes epithelial cell proliferation. In Hand2-knockout mice, the expression of FGF, ERK1/2, and Erα was up-regulated in the uterine epithelium [20]. This suggests that Hand2 is essential for mediating the downregulation of estrogen target genes. Furthermore, mitogen-induced gene-6 (MIG-6) is a downstream molecule of PGR. Based on the Mig-6^d/d^ and Mig-6^d/d^ErbB2^d/d^ mice models, MIG-6 inhibits epithelial cell proliferation by blocking ErbB2-ERK signaling [21]. However, MIG-6 expression was down-regulated in secretory endometrium in EMS patients.

Wnt signaling, specifically the Wnt/β-catenin pathway, has a significant role in activating implanted blastocysts, uterine development, and decidualization. During the decidualization of the mouse uterus, Wnt signaling dynamically expresses different Wnt ligands, frizzled (Fzd) receptors, inhibitors (i.e., Dkk1 and Sfrps), and transcriptional activators (e.g., β-catenin, etc.). Bone morphogenetic protein 2 (Bmp2), a member of the BMP morphogenetic subfamily, functions downstream of the PGR and is a key paracrine factor in transmitting embryonic adhesion signals from the epithelium to the stroma, which initiates decidualization. Using the Bmp2-conditional knockout mouse model, Bmp2 was proven to play a vital role in initiating decidualization by regulating FK506-binding proteins (Fkbps) and Wnt ligands. In addition, conditional knockout of uterine β-catenin in mice resulted in impaired endometrial decidualization [22]. The Bmp2-Wnt4/β-catenin pathway plays an important role in endometrial decidualization in mice and humans. However, abnormal expression of the Wnt family gene variant was observed in the endometrium or decidua of patients with EMS [23].

In addition, PR can interact with other nuclear transcription factors to regulate decidual processes. FOXO1, a member of the forkhead box O family (FOXO), is a nuclear transcription factor participating in the process of cell differentiation, apoptosis, and cell cycle control. It has been reported that FOXO1 can physically combine with PR to promote the expression of decidua-related genes insulin-like growth factor binding protein 1 (IGFBP1) and prolactin (PRL) [24]. As a result, the expression of FOXO1 mRNA in the secretory endometrium of patients with EMS was decreased as compared with normal endometrial tissue. Further studies have also found that overactive PI3K/AKT pathways in patients with EMS resulted in a down-regulated FOXO1 expression [25].

### 4.2. Estrogen Dominance

Estrogen is a key determinant in regulating the transition of the endometrium into a receptive state. Using the P4-treated delayed-implantation mice model, Wen-ge et al. found that the WOI duration was prolonged at lower estrogen concentrations but rapidly diminished at higher concentrations [26]. Hence, the levels of estrogen determine the duration of the receptive window in the uterus. EMS is an estrogen-dependent gynecological disease characterized by hyperactive estrogen signaling, resulting in the overproduction of 17-βestradiol (E2), which affects its function on the eutopic endometrium and ectopic foci. G. Anupa et al. also reported that dysregulated 17β-HSD1 expression resulted in hyperestrogenism rather than an anomaly in aromatase expression [27].

Estrogen promotes the transition of the endometrium to the secretory phase by interacting with estrogen receptors (ERs/ESRs) to induce mucosal proliferation and drive progesterone receptor synthesis during the proliferative phase. The ERs exist in two isoforms: ESR1/ERα and ESR2/ERβ. After binding to estrogen, ESR undergoes a conformational change and translocates into the nucleus, interacts with estrogen response elements or other transcription factors, and recruits coactivators to regulate the transcription of target genes [28]. In the normal endometrium, the expression of ERα is significantly higher than that of ERβ [29]. In ERα-knockout mice, the endometrium failed to support the blastocyst implantation. Thus, ERα is an important mediator of estrogen signaling in establishing a pregnancy. However, abnormal expression of ESR1 has been reported in EMS [30].

There are conflicting reports on the expression of endometrial ESR in patients with EMS. Most studies show that the eutopic endometrium of infertile patients with EMS exhibited notably higher ESR1/ERα levels during the secretory phase [31]. Over-expression of ERα during the mid-secretory phase negatively regulated integrin β3 in some women with EMS [29]. However, its exact molecular regulation mechanism still requires further research.

Hence, hormone imbalance leads to a series of changes in the eutopic endometrium, interfering with normal embryo implantation (Figure 1). Insights into the molecular basis of abnormal P4 and E2 signaling in the endometrium will help provide more targeted and individualized treatment options for patients with EMS-associated infertility.

## 5. Inflammation

At the maternal–fetal interface, maintaining an optimal pro- and anti-inflammatory status is essential for a successful pregnancy. Excessive inflammation is not conducive to embryo implantation. Peripheral blood (PB) and peritoneal fluid (PF) in patients with EMS contain a variety of products secreted by endometriotic implants and immune cells (e.g., growth factors, steroid hormones, inflammatory mediators), constructing an inflammatory microenvironment [32,33]. The characteristics of pro- and anti-inflammatory cytokines dynamically change in patients with EMS. As the disease continues to advance, the ratio of pro- and anti-inflammatory cytokines in PF gradually changes, characterized by a shift from Th1 (bad for pregnancy, e.g., interleukin-1β (IL-1β), interferon-γ (IFN-γ), tumor necrosis factor α (TNFα)) [34,35] to Th2 (good for pregnancy, e.g., IL-4, IL-10, transforming growth factor β (TGFβ)). Other studies have reported elevated levels of proinflammatory cytokines in the eutopic endometrium in patients with EMS [36]. Given the high clinical heterogeneity and differences in detection methods, the results of different studies are inconsistent. However, it is generally believed that the levels of TNF-α, IL-1, IL-6, and IL17 in the peritoneal fluid of patients with EMS are elevated [37]. Lessey et al. found that PF has a detrimental effect on the endometrium (LIF and integrin αvβ3) in women with EMS [38], suggesting that the inflammatory cytokines in the PF of EMS patients exert a direct influence on the endometrium. In addition, Llarena et al. examined the endometrial fluid of women with EMS to assess the endometrial microenvironment and found that the levels of proinflammatory cytokines (IL-1α, IL-1β, and IL-6) in the endometrial secretions of EMS patients were significantly higher [39]. Thus, EMS can impair fertility by increasing the concentration of proinflammatory cytokines (Figure 2). 

LIF is a member of the interleukin-6 (IL-6) family, which includes IL-6, LIF, IL-11, and so on. LIF regulates various cellular functions in autocrine and paracrine manners by combining with the LIF cell-membrane receptor, LIFR, and gp130 [40]. LIF has been demonstrated to play an important role in several processes of embryo implantation, including blastocyst growth and development, uterine preparation for implantation, and decidualization [41]. Dimitriadis et al. reported that the expressions of IL-11 and LIF decreased in the glandular epithelium of females with EMS, which was proven to be detrimental to pregnancy [42].

IL-1β plays an important role in the pathogenesis of EMS. Numerous studies have confirmed that the levels of IL-1β are significantly higher in the PB [43] and PF [44] of patients with EMS compared with control samples. Elevated levels of IL-1β in the endometrial fluid were negatively correlated with clinical pregnancy rates [45]. IL-1β can interfere with either the differentiation of endometrial stromal cells (ESCs) or embryonic implantation. Connexin43 (Cx43), a uterine gap junction protein, is critical for establishing cellular networking via the transfer of ions and signaling molecules between cells for embryonic implantation. IL-1β down-regulated the expression of Cx43, prolactin, IGFBP-1, and VEGF proteins via the ERK-MAPK pathway in vitro model of human ESC [46]. IL-1β also alters the gene expression of the endometrium. In baboon and mouse models, IL-1β dramatically down-regulated the expression of Homeobox A10 (HOXA10), HOXA11, and IGFBP1 [47].

IL-6 is considered a proinflammatory cytokine, involved in multiple processes such as the regulation of immune responses, inflammation, angiogenesis, and pregnancy outcome. IL-6 acts through gp130 after the activation of a specific IL-6 receptor (IL-6R). The main downstream signaling pathways mediating the action of IL-6 are the signal transducers and activators of transcription3 (STAT3) pathway and the mitogen-activated protein kinases (MAPKs) pathway. The molecular role of IL-6 is important for several biological processes that occur during pregnancy establishment, including trophoblast invasion, mediating immune tolerance for pregnancy, and placental development. There is growing evidence that the altered expression of IL6 ligands and/or their signaling regulators is associated with reproductive disorders. In EMS, IL-6 can be secreted by various cell types (peritoneum, macrophages, NK cells, endometrial epithelial cells, etc.) [48]. IL-6 expression is significantly higher in the peripheral blood [49], PF [50], and endometrial fluid [39].

TNF-α, a well-known member of the TNF superfamily, is synthesized by macrophages, T cells, natural killer cells (NKs), luminal epithelium, and decidual cells. TNF signals play important physiological and pathological roles in human endometrium and early implantation by regulating inflammation, apoptosis, proliferation, differentiation, and cell migration. However, over-expression of TNF-α is also detrimental to embryo implantation.

From this, it can be seen that EMS causes a local inflammatory microenvironment in the PF and endometrium, leading to impaired endometrial receptivity. Targeted inflammatory therapy to improve endometrial receptivity will have great potential in the future.

## 6. Endometrial Immunoregulatory Dysfunction

EMS is widely considered an estrogen-dependent inflammatory disease. Significant immune cells (e.g., NK cells, Treg, macrophages, and B cells) are recruited into the pelvis and eutopic endometrium. Endometrial immune cells secreting chemokines and cytokines are key participants in the cell–cell communication pathways that regulate endometrial receptivity to embryo implantation. Increasing evidence supports that the altered immune status of eutopic endometrium in patients with EMS is consistent with that in the peritoneal environment, which likely contributes to infertility and early pregnancy failure (Figure 3) [51,52]. Transcriptome meta-analysis reveals that a pro-inflammatory profile was predominant in eutopic endometrium from stage I-II EMS [53].

### 6.1. Macrophages

Macrophages (Mφ) are key effector cells on the front line of innate and humoral immunity. Mφ have two activation pathways: “classically activated” M1 and “alternatively activated” M2. M1 plays a role in proinflammatory responses and secretes cytokines (IL-1, IL-6, IL-8, prostaglandin E2 (PGE2), hepatocyte growth factor (HGF), IL-12, IL-23, nitric oxide synthase (NOS)), whereas M2 is involved in anti-inflammatory reactions, angiogenesis, and tissue repair. Mφ are essential for successful implantation. Yosuke et al. found that in the CD206 diphtheria-toxin-receptor mouse model, aberrant expression of fibroblast growth factor18 (FGF18) promoted endometrial epithelial cell proliferation, which led to implantation failure [54]. Ding et al. observed that M2-derived growth-colony stimulating factor (G-CSF) promoted the epithelial-to-mesenchymal transition in trophoblasts, both invasion and migration while mediating normal pregnancy through activating PI3K/Akt/Erk1/2 signaling [55]. In animal models, Mφ with a TIE2^+^ proangiogenic phenotype have an essential role in supporting corpus luteum formation and progesterone synthesis in the peri-implantation period [56]. Macrophage phenotypes and functions in the endometrium change with the menstrual cycle. 

In the healthy endometrium, the predominant macrophage phenotype is Mφ2, suggesting that the normal environment is anti-inflammatory. Nonetheless, limited research has been carried out in exploring the phenotypic character of Mφ in the endometrium of women with EMS. Khan et al. reported that the infiltration of Mφ was significantly decreased in the eutopic endometrium and ectopic foci of women with mild EMS than that with advanced EMS or in the control samples [57]. Akie et al. reported that the ratios of Mφ2 were significantly lower at all phases in the eutopic endometrium of EMS patients [58]. Linda et al. showed that Mφ displayed a higher proinflammatory phenotype in eutopic endometrium of EMS patients compared to the control samples without any disease [59].

### 6.2. NK Cells

NK cells are the most abundant type of lymphocytes in the endometrium and play an important role in the female reproductive system. NK cells change dynamically with the menstrual cycle, accounting for 30–40% of total leukocytes (proliferative phase) and up to 70% (secretory phase). Uterine natural killer (uNK) cells are considered the foundation of successful embryo implantation and pregnancy; however, the origin of the uNK cells remains unclear. Brighton et al. found that uNK cells play a pivotal role in embryo biosensing and determining endometrial fate decisions during implantation [60]. The predominant uNK cells are characterized as CD56^bright^CD16^−^. Low cytotoxic CD56^bright^CD16^−^ NK cells can secrete angiogenic factors (VEGF) that promote the remodeling of the spiral artery and cytokines (e.g., C-X-C motif chemokine 10 (CXCL10), CXL12, colony-stimulating factor 1 (CSF-1), transforming growth factor-β (TGFβ), IL17, and LIF), which may direct the migration and invasion of the trophoblast.

Dysfunctional NK cells are related to infertility. Giuliani et al. reported that patients with EMS have a larger amount of cytotoxic CD16^+^ uNK cells and may suffer a higher risk of infertility resulting from an inflammatory environment that is detrimental to implantation or decidualization [61]. In the primate model of EMS, eutopic uNK (NKp30 expressing) cells and immuno-expression for BAG6 increase in the late-secretory phase, particularly at the very early stages of EMS [62]. Bcl-2-associated athanogene 6 (BAG6) serves as the ligand for NKp30, triggering NK cell cytotoxicity [63].

### 6.3. T Cells

T cells function as the main components of lymphocytes which mediate cellular immunity and also regulate the immune function of the body. Based on their phenotype and function, mature T cells are mainly divided into three subtypes: cytotoxic T lymphocytes (CD8^+^T), T helper cells (CD4^+^T), and regulatory T cells (Treg). Tregs via and in response to cytokines play crucial roles in regulating maternal–fetal tolerance to implantation and pregnancy. Therefore, a reduction in the amount and dysfunction of Treg in the endometrium leads to implantation failure. Several studies found that FoxP3^+^ Tregs fluctuate with the menstrual cycle and are reduced in infertile patients with mild EMS (r-ASRM stages I and II) [64,65].

### 6.4. Dendritic Cells

Dendritic cells (DCs) are powerful antigen-presenting cells (APCs) that are highly involved in T-cell stimulation and the maintenance of immune tolerance. There are two different states of DCs in the maternal uterus: mature dendritic cells (CD83^+^HLA-DR^high^ mDCs) and immature dendritic cells (CD1a^+^CD209^+^HLA-DR^low^ iDCs). The main cells that regulate maternal immune tolerance are iDCs. DC populations are less common than other types of leukocytes during the normal menstrual cycle. iDCs fluctuate in density with the menstrual cycle in the basal layer of the endometrium, while the populations of mDCs remain controversial. Notably, iDCs do not express estrogen or progesterone receptors but are periodically regulated by hormones. DC populations are altered in the eutopic endometrium of patients with EMS. Schulke et al. reported that the endometrial density of CD1a^+^ iDCs increased in the proliferative phase, but CD83^+^ mDCs decreased across all phases in women with EMS compared with the control samples [66]. Further analysis was performed at each EMS disease stage with local and circulating DC subtypes. McGuire et al. showed that the endometrial density of CD141^+^ mDC reduced and IRF-8^+^ increased in advanced EMS (Stages III–IV), whereas menstrual cyclical fluctuations in CD1c^+^ and IRF-8^+^ DCs decreased in EMS [67]. 

Several researchers reported that DC dysfunction led to reproductive disorders including implantation failure [68,69]. The exact mechanism of DCs during the peri-implantation remains unclear; however, DC-derived soluble factors, which include matrix metalloproteinases (MMPs), cytokines (IL-12, TNF-α, IL-8), and chemokines (macrophage inflammatory protein-1α (MIP-1α), MIP-3α, regulated on activation in normal T-cell expressed and secreted (RANTES), interferon-inducible protein-10 (IP-10)), are likely to affect endometrial local immune microenvironment during implantation [70]. 

Therefore, it is important to explore the characteristics of immune cells in PF and endometrium of EMS patients with different stages. Clarifying the mechanisms by which immune dysfunction interferes with endometrial homeostasis facilitates the search for new treatments.

## 7. Epigenetic Dysregulation

Epigenetic alterations can alter gene expression and function by reversible epigenetic modification. Epigenetics phenomena mainly include DNA methylation, histone modifications, non-coding RNA regulation, and genomic imprinting. Recently, many studies have provided clues to reveal that epigenetic aberrations affect reproduction in the regulation of implantation, decidualization, and fetal growth [71]. Abnormal epigenetic modifications of several genes have also been found in women with EMS as well as in EMS animal models [72,73].

### 7.1. DNA Methylation

DNA methylation is a post-synthetic modification of DNA that frequently results in gene silencing. In EMS, endometrial receptivity is epigenetically regulated. E-cadherin is the primary adhesion molecule involved in embryo implantation. Rahnama et al. reported that DNA methylation decreases the E-cadherin expression during trophoblast invasion [74]. HOXA10 is necessary for embryo implantation, which is down-regulated in the mid-secretory endometrium of women with EMS. Guo et al. demonstrated that HOXA10 was hypermethylated in the endometrium of women with EMS [75]. Furthermore, PGR-B and ESR2 are methylated in the eutopic endometrium of women with EMS-related infertility, which impairs endometrial receptivity [76].

### 7.2. Histone Acetylation

Histone acetylation has been reported to induce endometrial pathologies such as EMS, endometrial cancer, and implantation failures. Histological evaluations have revealed aberrant histone acetylation in the eutopic endometrium of patients with endometriosis. Histone acetylases (HATs) and histone deacetylases (HDACs) play key roles in regulating gene expression by changing the chromatin structure. HDACs, as transcriptional repressors, generally hinder gene expression. Kim et al. found that the expression of HDAC3 protein was reduced in the eutopic endometrium of patients with EMS, while in vivo and in vitro studies also demonstrated that HDAC3 loss contributed to the expression of collagen type I alpha 1 (COL1A1) and COL1A2 genes and ultimately caused implantation and decidualization defects [77]. Maryam et al. reported that the reduction in HOXA10 expression was associated with lower acetylation and higher methylation of H3K9 in secretory endometrium in women with EMS [72].

Epigenetic Dysregulation impairs endometrial receptivity in EMS. Thus, whether targeted epigenetic therapy can improve endometrial tolerance is a clinical question for future exploration.

## 8. Glycosylation

Protein glycosylation is the covalent attachment of a glycan group (mono-, di-, or polysaccharide) to select residues of target proteins. According to distinct protein–sugar linkages, glycosylation involves N-glycosylation (sequon: Asn-X-Ser/Thr, X stands for any amino acid), O-glycosylation (sequon: Ser/Thr), C-glycosylation (sequon: Trp-X-X-Trp), S-glycosylation (sequon: Cys), and P-glycosylation (sequon: Ser/Thr). As a complex post-translational modification, protein glycosylation deeply affects the correct localization, conformational stability, and function of membrane proteins. The types of glycosylation or subtleties of oligosaccharide structure are key role determinants in regulating cell adhesive interactions, enzyme activity, and protein phosphorylation states. The endometrium transforms into a receptive state for embryo implantation, experiencing substantial remodeling of the glycoprotein network in the endometrial epithelium. Each species has its own unique pattern of glycosylation, which is important in maternal–fetal recognition and the establishment of pregnancy. Histological evaluations have reported that many different oligosaccharide structures exist at the human maternal–fetal interface and change dynamically during the embryo implantation process [78]. Studies linking cycle stage-dependent glycosylation to specific glycoproteins expressed during WOI, such as MUC-1 and Glycodelin-A (GdA), have shown changes in the expression profile, possibly associated with impaired endometrial receptivity in EMS [79]. In addition, histological evaluations showed a significant reduction in Dolichos biflorus agglutinin-binding glycans in the peri-implantation phase of the endometrium in women with advanced EMS [80]. 

Yet so far, the mechanism of glycosylation regulation in human endometrial epithelium remains unclear. Many laboratory studies have demonstrated that the expressions of glycosyltransferases such as fucosyltransferase 2 (Fut2), Fut4, and ST3 beta-galactoside alpha-2,3-sialyltransferase 3 (ST3Gal3) were significantly increased during implantation [81,82]. Thus, glycosyltransferase activity may be the main factor accounting for this abnormal response. Studies have demonstrated that estrogen and progesterone can modulate the expression of glycosyltransferases [83]. Immune cells (such as macrophages, NK, and Treg cells) play a crucial role in the establishment of pregnancy. Sophia et al. demonstrated that disruption of the NK cell–DC dynamics directly affects the placental glycocode, leading to placental dysfunction [84]. It is postulated that in EMS, abnormal immune cells in the eutopic endometrium may influence the cytokine profiles at the implantation site and directly affect the expression of glycosyltransferases to alter the expression of adhesion molecules in the glandular epithelial.

## 9. Metabolites and Metabolic Enzymes

Intracellular metabolism is closely related to its phenotype and function. Glucose, amino acids, and fatty acids are the three major energy sources for maintaining cell life activities. They generate energy and biologically active metabolites and participate in physiological processes such as cell proliferation, differentiation, activation, and apoptosis. Metabolites are not only the final products of the physiological regulatory process but also act as mediators to regulate the body’s homeostasis. Lipids play an important role in the development of many pathological processes, such as inflammation, oxidative stress, and angiogenesis, all of which are related to the pathogenesis of EMS. There is no doubt that EMS causes metabolic changes. Recently, several studies have revealed metabolic characteristics of the endometrium [85], endometrial fluid [86], and serum [87] of patients in different stages of EMS (Table 2). These studies suggest that endometrial lipid metabolism is abnormal in patients with EMS. 

### 9.1. Phospholipids

Phospholipids are amphiphilic molecules grouped into two categories: glycerophospholipids and sphingomyelins, which are composed of glycerol and sphingosine, respectively. Phospholipids, as structural components of cell membranes, are the precursors of various messenger molecules and are involved in signal transduction. The correlation between embryo implantation and metabolites from phospholipid metabolism has been extensively studied. Phospholipid-derived mediators, such as endocannabinoids, lysophosphatidic acid (LAP), and sphingosine-1-phosphate (SIP), have been implicated in endometrial receptivity and decidualization. Uterine angiogenesis coordination has been suggested as a potential mechanism by which SIP signaling promotes decidualization. SIP can induce COX-2 expression in stromal cells, suggesting a link between sphingolipids and PG signaling during early pregnancy. Sphingolipid monohexosylceramide and Ceramides (Cers) were found to be down-regulated in patients with ovarian EMS [89]. Considering that sphingolipids and Cers are involved in various cellular processes (such as proliferation, migration, and apoptosis), the decreased expression of such metabolites may affect embryo implantation [89].

### 9.2. LPA-LPA Receptor (LPA3) Signaling

Lysophosphatidic acid (LPA), as a small molecule lipid signal, participates in a variety of physiological and pathological processes, including cell proliferation, differentiation, cytoskeleton arrangement, and cell invasion. LPA3, a G protein-coupled receptor, is the major subtype of LPA receptor involved in embryo implantation. LPA3-knockout female mice exhibited delayed embryo implantation and altered embryo spacing through the down-regulation of prostaglandins expression levels [91]. Sordelli et al. found that LPA-LPA3 signaling was involved in vascular remodeling and ensured decidualization and placenta formation at the maternal–fetal interface [92]. However, immunohistochemical studies showed that the LPA3 expression was decreased in the mid- and late-secretory endometrium in women with EMS [93].

### 9.3. Glucose

Glucose provides an energy substance for the endometrium, which is essential not only for the growth of the embryo in the uterus but also for the transformation of the endometrium into a receptive state at the time of embryo implantation. The endometrium undergoes a series of changes such as differentiation, cytoskeleton remodeling, secretion, and resorption of uterine fluid, all of which require energy expenditure. Therefore, adequate glucose uptake and metabolism by endometrial cells are critical for establishing endometrial receptivity. GLUT protein is the main pathway for glucose to enter cells. GLUT4 is one of the glucose transporters, which is localized in the epithelial cell membrane and is strongly expressed during WOI. GLUT4 may provide an important pathway for glucose to enter the endometrial epithelial cells and participate in the maintenance of glucose concentration in the uterine fluid. Recent studies found that the expression of GLUT4 in the eutopic endometrium of EMS patients was decreased. Thus, the endometrial glucose uptake disorder in patients with endometriosis may be one of the reasons for the decreased endometrial receptivity [94].

Based on evidence from animal and human studies, metabolic disorders may reduce endometrial receptivity [95,96]. Several lipid molecules have been reported to take part in regulating embryo implantation. Metabolites have also been considered as a potential marker for assessing endometrial receptivity. However, there is limited information on the metabolic characteristics of the endometrium during WOI in EMS. Therefore, it is of great significance to deeply explore the metabolic characteristics during WOI and their physiological and pathological roles in patients with EMS.

## 10. MicroRNAs Dysregulation

MicroRNAs (miRNAs) are RNA molecules consisting of 21–23 nucleotides and function as post-transcriptional regulators of gene expression. miRNAs are involved in a variety of physiological mechanisms, which are key regulators of the development and maintenance of intracellular homeostasis. Kuokkannen et al. reported that there were significant differences in miRNA expression between proliferative and secretory endometrial tissues, suggesting that miRNA expression was regulated by estrogen and progesterone [97]. With the discovery of extracellular miRNAs, recent studies identified that miRNAs can act as mediators of maternal–fetal dialogue and participate in the establishment of endometrial tolerance [98]. Thus, miRNA dysregulation affects endometrial receptivity. Currently, alterations in miRNA expression have been reported in the eutopic endometrium of EMS patients. Microarray profiling revealed that the expression of miR-543 was down-regulated in the endometrium of patients with EMS-associated infertility during WOI, which may affect embryo implantation [99]. Petracco et al. found that miR135a and miR135b were upregulated in the endometrium of EMS patients, further demonstrating that the over-expression of miR135 down-regulated the expression of implantation-related genes, including HOXA10 [100]. Joshi et al. found that the increased expression of miR-29c in the endometrium of EMS patients can lead to impaired expression of FKBP4, which may be one of the mechanisms leading to progesterone resistance [101]. Pei et al. found that increased expression of miR-194-3p may be associated with the inhibition of ESC decidualization in vitro [102].

Thus, current studies suggest that endometrial miRNA abnormalities in EMS patients affect processes such as embryo implantation and decidualization, despite the exact role they play being unclear.

## 11. Intervention Strategies to Improve Endometrial Receptivity

### 11.1. Surgery

As previously mentioned, numerous experimental studies have shown that an abnormal pelvic inflammatory microenvironment in patients with EMS may impair endometrial receptivity. Surgical resection of ectopic lesions is proposed to improve endometrial receptivity. However, there are no current clinical data showing that surgical treatment of EMS can improve endometrial receptivity. Mikhaleva et al. performed endometrial biopsies on EMS patients, who underwent surgical resection of the ectopic ovarian foci after 6 and 12 months and found that the ER and PR were extensively expressed in the proliferative and secretory endometrium, with the impaired formation of pinopodia in the secretory phase [103]. Thus, surgery might not be an option to overcome the bio-molecular alterations of the endometrium due to EMS.

### 11.2. Metformin

Metformin, a widely used anti-diabetic drug, is known for its multi-purpose application for other disease treatments. In recent years, metformin has been applied as a treatment option for endometriosis-related infertility. A clinical trial study found that metformin improved pregnancy rates with a 0% to 25.7% chance in patients with EMS [104]. Studies involving in vitro and animal models also demonstrate the potential of metformin in improving endometrial receptivity in EMS. Zhou et al. reported that metformin hindered PGE2-induced CYP19A1 mRNA expression and aromatase activity in the endometriotic stromal cells by inhibiting CREB binding to the promoter (PII) [105]. Cheng et al. demonstrated that metformin increased the expressions of LIF and HOXA10 in the endometrium of EMS mice [106]. Furthermore, studies have shown that metformin can suppress inflammation, which plays an important role in the development of EMS and is mainly related to decreased endometrial receptivity. Therefore, metformin provides a promising potential for EMS treatment by inhibiting the release of IL-6 and IL-8, which reduces inflammation [107].

### 11.3. Aromatase Inhibitors

Aromatase p450 is a key enzyme in the synthesis of estradiol. Expression of Aromatase P450 was higher in the eutopic endometrium in patients with EMS compared with normal endometrium [108]. Aberrantly expressed aromatase can cause local elevation of estrogen levels in the endometrium. Letrozole is mainly known for its clinical application as an aromatase inhibitor. Due to its ability to inhibit estrogen biosynthesis, letrozole is a potential drug that could correct abnormal endocrine and reproductive functions of endometriosis. Based on the mouse model of EMS, Zhang et al. found that letrozole could increase the expression levels of endometrial integrin αvβ3 and HOXA10 in mice to improve endometrial receptivity, but its specific mechanism still warrants further experimentation [109]. Limited clinical research evidence suggests that letrozole may potentially improve endometrial receptivity in EMS patients undergoing in vitro fertilization (IVF) [110].

### 11.4. Immunomodulation Therapy

Given the important role of immunological mechanisms in EMS, immunomodulatory therapies for EMS-associated infertility have been explored in animal and clinical studies. However, studies remain scarce. Pentoxifylline is an anti-inflammatory (including down-regulation of TNF-α) and immunomodulatory agent recommended for the treatment of EMS. Animal studies showed that pentoxifylline remarkably inhibited the growth of ectopic lesions [111]. Yet, a recent Cochrane systematic review reported that there was no sufficient evidence to support that pentoxifylline could alleviate pain or improve the pregnancy rate in EMS patients [112]. TNF-α antagonists include recombinant TNFRSF1A (soluble form of TNF receptor type 1), etanercept, and infliximab. However, only animal studies reported that TNF-α antagonists could significantly inhibit the growth of ectopic foci and improve pain symptoms [113]. So far, there is still no high-level clinical evidence to support the use of TNF-α antagonists to improve the fertility of EMS patients [114]. In addition, stem cell therapy is a potential immunomodulatory strategy to improve endometrial function [115,116]. Actually, no well-designed studies evaluated the clinical value of stem cell therapy in improving endometrial receptivity.

In recent years, as the research on immune-induced EMS-related infertility continues to advance, so does the corresponding immunomodulatory therapy. The use of immunomodulatory drugs currently has certain limitations, such as adverse effects and embryotoxicity, which need to be explored in depth using more prospective studies.

### 11.5. Others

Protopanaxadiol (PPD) is a chemical monomer with an active ingredient called ginsenoside, and its chemical structure is similar to that of steroid hormones. Lai et al. found that PPD may improve endometrial receptivity by inhibiting the expression of proinflammatory cytokines by binding to intracellular receptors/sensing proteins of macrophages without embryotoxicity [117]. Therefore, PPD may serve as a potential drug for improving fertility in EMS patients.

## 12. Conclusions and Future Perspectives

In summary, we have outlined the factors of EMS that may affect endometrial receptivity (Figure 4). With the development of assisted reproductive technologies, endometrial receptivity will be a bottleneck for a successful pregnancy. However, there are still few methods to assess endometrial receptivity or available therapeutic drugs. EMS is a highly heterogeneous, multisystem, complex gynecological disorder that causes pain and infertility. Inflammation is the basis for EMS-related symptoms. The complex and interconnected pathways in the endometrial microenvironment link inflammation with metabolic and immune responses; thus, a balance between inflammation and metabolic and immune homeostasis is particularly important to maintain endometrial function. In fact, the continued deepening of emerging immunometabolism research will certainly further reveal its role in endometrial physiology and metabolism, possibly revealing novel diagnostics, models, and therapeutic approaches for EMS-associated subfertility/infertility.

## Figures and Tables

**Figure 1 biomolecules-13-00430-f001:**
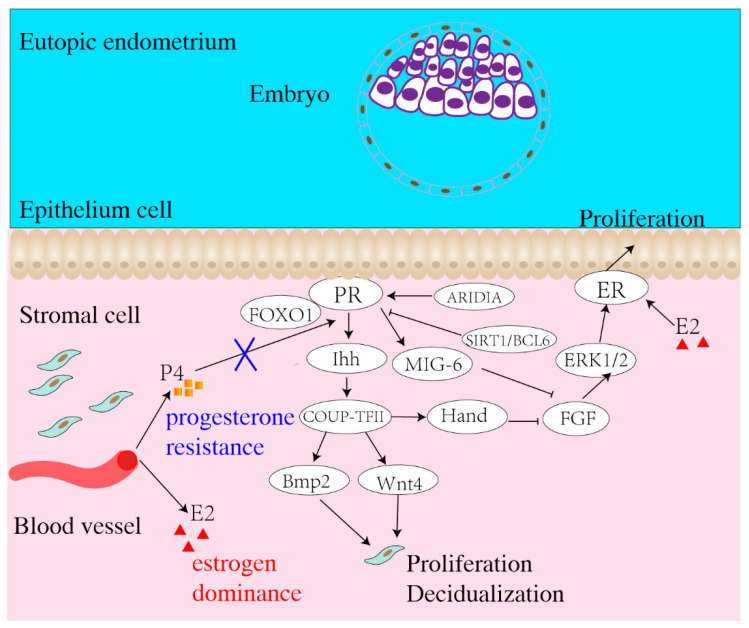
Schematic illustration of the mechanism of progesterone and estrogen signaling in the regulation of endometrial receptivity. In the normal endometrium, progesterone and estrogen signaling work together to inhibit epithelial proliferation and facilitate the transition to an embryo-receptive state during the window of implantation. Yet in EMS, the dysfunctions of progesterone and estrogen signaling bring about progesterone resistance and estrogen dominance. ARIDIA, AT-rich interacting domain protein 1A; BCL6, B-cell lymphoma 6; BMP2, bone morphogenetic protein 2; COUP-TFII, chicken ovalbumin upstream transcription factor II; E2, estrogen; ER, estrogen receptor; ERK, extracellular signal-regulated kinase; FGF, fibroblast growth factor; FOXO1, forkhead box O; HAND2, heart- and neural crest derivatives-expressed 2; IHH, Indian Hedgehog; MIG-6, mitogen-induced gene-6; P4, progesterone; PR, progesterone receptor; SIRT1, sirtuin1; Wnt4, wingless-related MMTV integration site4.

**Figure 2 biomolecules-13-00430-f002:**
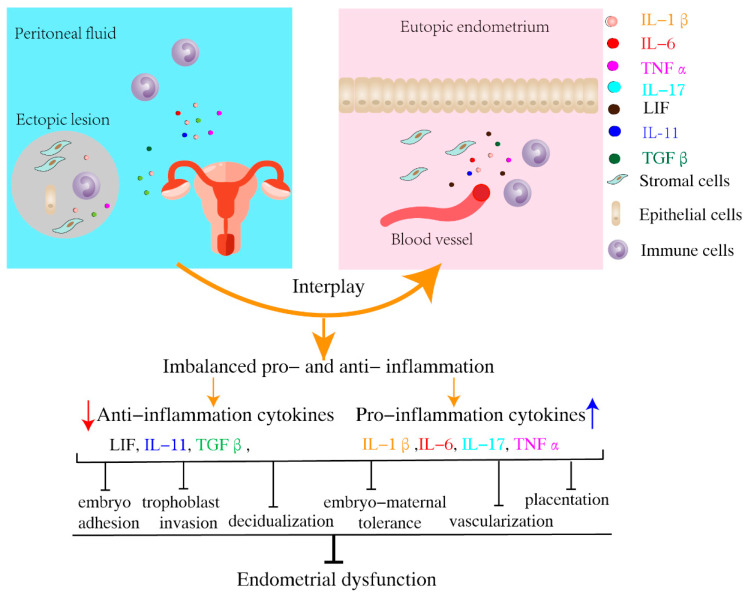
The effects of inflammation on endometrium in women with EMS. EMS, endometriosis; IL, interleukin; LIF, leukemia inhibitory factor; TGF, transforming growth factor; TNF, tumor necrosis factor.

**Figure 3 biomolecules-13-00430-f003:**
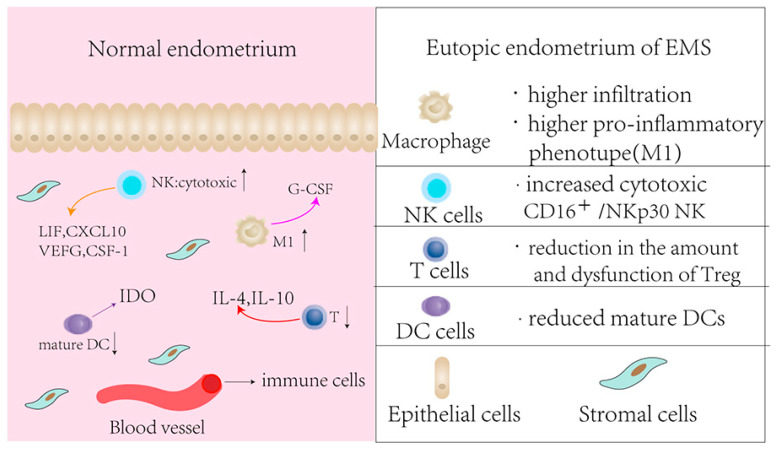
Schematic presentation of the endometrial immune environment of women with EMS. CSF, colony-stimulating factor; CXCL, chemokine (C-X-C motif) ligand; DC, dendritic cell; EMS, endometriosis; G-CSF, granulocyte-colony stimulating factor; IDO, indoleamine 23 dioxygenase; IL, interleukin; LIF, leukemia inhibitory factor; NK, natural killer; VEGF, vascular endothelial growth factor; TGF, transforming growth factor; TNF, tumor necrosis factor.

**Figure 4 biomolecules-13-00430-f004:**
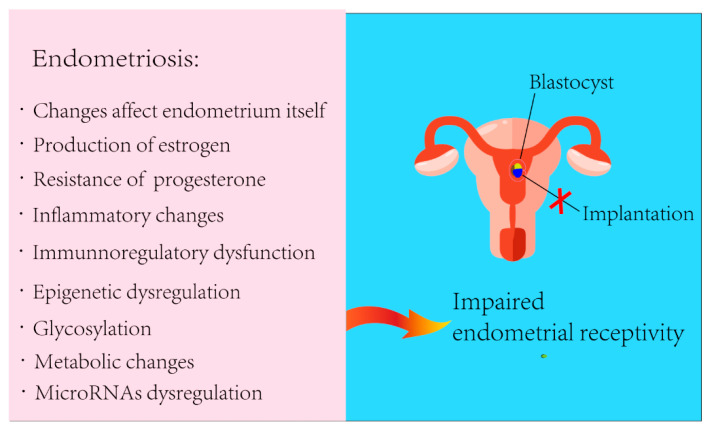
A summary of the pathogenesis of EMS affecting endometrial receptivity.

**Table 1 biomolecules-13-00430-t001:** Gene profile and protein expression of secretory endometrium in EMS.

Reference	Sample Number	Analysis Platform	Main Findings
[10]	n = 36: EMS II (n = 6); EMS III (n = 6); EMS IV (n = 6); Controls (n = 18)	proteomic analysis	The differentially expressed proteins involved in stress response, protein-folding and protein-turnover, immunity, energy production, signal transduction, RNA biogenesis, and protein biosynthesis.
[11]	n = 20: mild-moderate EMS (n = 8); Controls (n = 12)	Affymetrix Genechip, microarray analysis	Dysregulation of selected genes involved in embryonic attachment, embryotoxicity, immune dysfunction, and apoptotic responses, aromatase, progesterone receptor, angiogenic factors.
[12]	n = 6: EMS (n = 3); Controls (n = 3)	lncRNA microarray; DNA microarray	Differently expressed lncRNAs associated with cell cycle and immune regulation.
[13]	n = 7: moderate-severe EMS (n = 4); Controls (n = 3)	miRNA microarray analysis	Differentially expressed mRNA genes associated with the biological processes of cell death, cell cycle, and cellular assembly and organization.
[4]	n = 37: mild-moderate EMS (n = 21); Controls (n = 16)	microarray analysis	Up-regulated genes are involved in the immune (GZMA, C4BPA) or inflammatory responses.
[5]	n = 16: EMS III-IV (n = 8); Controls (n = 8)	immune and inflammation transcriptomic analysis	Significantly different genes involved in the regulation of cell apoptosis and decidualization.

**Table 2 biomolecules-13-00430-t002:** Characteristics of endometrium metabolomic studies in patients with EMS.

Reference	Sample Number	The Phase of the Menstrual Cycle	Analysis Method	Sample Source	Main Findings
[86]	n = 35: EMS (n = 12); Controls (n = 13)	the window of implantation	ultrahigh performance liquid chromatography coupled to mass spectrometry (UPLC-MS)	Endometrial fluid (EF)	Glycerolipids and glycerophospholipids were overrepresented in the EF of women with EMS. The sphingolipids CMH and ceramidesin levels were lower in the EF of women with EMS.
[88]	n = 66: EMS (n = 29); infertile women (n = 37)	on the third to fifth day after menstrual cessation	ultra-high-performance liquid chromatography coupled with electrospray ionization high-resolution mass spectrometry (UHPLC-ESI-HRMS)	Eutopic endometrium	The eutopic endometrium metabolomic profile of the EMS patients was characterized by a significant increase in the concentration of hypoxanthine, L-arginine, L-tyrosine, leucine, lysine, inosine, omega-3 arachidonic acid, guanosine, xanthosine, lysophosphatidylethanolamine, and asparagine.
[87]	n = 119: EMS I (n = 20); EMS II (n = 13); EMS III (n = 17); EMS IV (n = 45); Controls (n = 24)	not clear	1H NMR spectra	Eutopic endometrium and Blood	A number of amino acids, including alanine, lysine, phenylalanine, and leucine, showed significantly lower levels in the endometrial tissue of women with EMS relative to healthy controls.
[89]	n = 12: EMS (n = 6); Controls (n = 6)	proliferative and secretory	LC-MS/MS	Eutopic endometrium	Upregulation of specific sphingolipid enzymes, namely sphingomyelin synthase 1 (SMS1), sphingomyelinase 3 (SMPD3), and glucosylceramide synthase (GCS) in the endometrium of EMS women with corresponding increased GlcCer levels and decreased sphingomyelin levels.
[90]	n = 41: EMS I-II (n = 21); infertile women (n = 20)	not clear	UHPLC-ESI-HRMS	Eutopic endometrium	Lipid profiles of early-stage (I–II) EMS patients were characterized by a decreased concentration of phosphatidylcholine and phosphatidylserine and an increased concentration of phosphatidic acid compared with the control sample.

## Data Availability

Not applicable.

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
