# Peer review of "Towards a Better Understanding of Endometriosis-Related Infertility: A Review on How Endometriosis Affects Endometrial Receptivity"

_biomolecules, 2023, doi:10.3390/biom13030430_

Round 1

Reviewer 1 Report

Congratulation This is an interesting well done review describing all the possible endometrial factors reducing embryo’s implantation.

However

·       Implantation failure is mainly due to the embryos aneuploidy which are about half embryos obtained in ART until 35 years old and then decrease according to age  (Pirtea P et al , 2021) until 17% in the 42 years old women . The embryo transfer of 3 euploid embryo leave the RIF as marginal phenomenon of 5% of women without have a baby . Thus the interest of this review is mainly for half of failure per euploid embryo transferred

·       In ART, oocyte quality and quantity ( when endometriosis is not including ovarian cysts ) does not change in patients with endometriosis as well as the % of euploid embryos( D De Ziegler et al ,2018) Thus the % of failure is due to what the authors are described in this review ( D DeZiegler et al 2010, 2018)

I believe that the authors should add a clinical value to their effort in the interest of the readers .

I suggest to have a review of the drop between embryo transfer and successful implantation according to age class and diagnosis of sterility within the national registers of ART in US , UK, Australia, Germany and South America because in that registers the data are furnished from single ART center as disaggregated. And then  to calculate the implantation rate of patients with endometriosis versus the other diagnosis . Much better if completed with report for those who transfer only euploid embryos. That would establish the new threshold for good standard effort in ART cycles for patients with endometriosis. Establishing the gold standard implantation rate specifically for patients with endometriosis.

Your  extensive review should then cover that % of difference that those patients have in comparison with the implantation rate of women undergoing ART cycles for other causes. Furthermore the efforts of the front of scientists that both are working on infertility centers or endometriosis research should have the privilege to read in that review the present efficiency of ART in endometriotic patients as well as the factors that may counteract the  detrimental effects of endometriosis to improve the reproductive efficiency of ART .

Diagnosis of

A.     Titolo 2

·       de Ziegler D, Borghese B, Chapron C. Endometriosis and infertility: pathophysiology and management. Lancet. 2010 Aug 28;376(9742):730-8. doi: 10.1016/S0140-6736(10)60490-4. PMID: 20801404.

·       de Ziegler D, Pirtea P, Carbonnel M, Poulain M, Cicinelli E, Bulletti C, Kostaras K, Kontopoulos G, Keefe D, Ayoubi JM. Assisted reproduction in endometriosis. Best Pract Res Clin Endocrinol Metab. 2019 Feb;33(1):47-59. doi: 10.1016/j.beem.2018.10.001. Epub 2018 Nov 3. PMID: 30503728.

·       Pirtea P, De Ziegler D, Tao X, Sun L, Zhan Y, Ayoubi JM, Seli E, Franasiak JM, Scott RT Jr. Rate of true recurrent implantation failure is low: results of three successive frozen euploid single embryo transfers. Fertil Steril. 2021 Jan;115(1):45-53. doi: 10.1016/j.fertnstert.2020.07.002. Epub 2020 Oct 16. PMID: 33077239.

Reviewer 2 Report

Thank you very much for the invitation to review of the manuscript. It a great pleasure for me.

The purpose of Shan et al. was to analyse a role of hormone imbalance, inflammation and immune regulatory dysfunction and also genetic, epigenetic, glycosylation and metabolism in endometriosis-related infertility/subfertility and their clinical implications in the diagnosis and therapy. This is very interesting paper, However, I have a few suggestions and advice:

1.     The methodology of the paper is unclear and the Authors should include more precise information about inclusion/exclusion criteria as well as basic key words used in literature research.

2.     In “introduction” line 30 – what about immune imbalance and chronic inflammation?

3.     All abbreviations should be explained

4.     How important is the immune and inflammation status of the peritoneal cavity environment on the implantation process? What differences in ectopic and ectopic tissue are crucial here?

5.     It is worth to add more information about role of LIF in implantation.

6.     It is worth to add more information about TGF-beta and fibrosis as one of the potential causes of infertility in women with endometriosis.

7.     do patients with endometrioma have hormone secretion disorders, and does removal of this lesion improve this?

8.     Have any trials been made with biological treatment? What about pentoxifylline or anakinra?

9.     The manuscript contains a lot of information, but sometimes it is very general.

10.  Additionally, quite often, I don’t know if you mean eutopic or ectopic endometrium or endometriosis lesion?

Reviewer 3 Report

The manuscript entitled " Towards a Better Understanding Endometriosis-Related Infertility: A review on How Endometriosis Affects Endometrial Receptivity " has provided a wide spectrum of topics related to endometriosis-promoted infertility state such as underlying mechanisms, hormonal dysregulations, immunological dysregulations, metabolites, genetic and epigenetic alterations, and methods to ameliorate endometriosis-caused infertility.

• The manuscript subject seems to be too general to cover all the findings regarding EMS infertility; as hypothalamic-pituitary-ovarian axis and ovarian function (doi: 10.1034/j.1600-0897.2003.01156.x), sperm-related infertility, cumulus cells (doi: 10.1007/s10815-016-0727-z) and the endometrial microenvironment (doi: 10.1016/j.rbmo.2013.06.010) are other aspects of endometriosis-related infertility not discussed in the manuscript.

• In the discussion section, the manuscript should point out the current limitations of therapy, and how the manuscript findings and conclusions can contribute to this scientific niche as well.

The following sections of the manuscript can also be improved:

1. For instance, the " improve endometrial receptivity " section can be enhanced by adding the related data involving GnRH agonists, progestins, surgical treatment alternatives (doi: 10.3390/jcm9020507), assisted reproductive technology (ART), and IVF.

2. The role of miRNAs in infertility caused by endometriosis can also be added: (DOI: 10.1210/en.2018-00374, DOI: 10.1177/1933719112453507, DOI: 10.1111/jog.15526). Additionally, the potential of mesenchymal stem cells as therapeutics can be looked at.

3. It is suggested to add a section devoted to the method of searching, utilized databases, and keywords.

4. A conclusion or summary at the end of each section can be beneficial.

5. The quality of the figures can be improved. For instance, inflammation-related agents can be illustrated with more distinctive colors and shapes in Figure 2. The provided figures could also be of better quality.

6. The abbreviations section should be added as well.

Round 2

Reviewer 2 Report

Thanks ror replay and take under consideration my suggestions. S

Reviewer 3 Report

Considering the current corrections of the article, I suggest its publication in the current format.